# Plants in the Genus *Tephrosia*: Valuable Resources for Botanical Insecticides

**DOI:** 10.3390/insects11100721

**Published:** 2020-10-21

**Authors:** Peiwen Zhang, Deqiang Qin, Jianjun Chen, Zhixiang Zhang

**Affiliations:** 1A Key Laboratory of Natural Pesticide and Chemical Biology, Ministry of Education, South China Agricultural University, Guangzhou 510642, China; peiwen.zhang@ufl.edu (P.Z.); 20181023002@stu.scau.edu.cn (D.Q.); 2Department of Environmental Horticulture and Mid-Florida Research and Education Center, Institute of Food and Agricultural Sciences, University of Florida, Apopka, FL 32703, USA

**Keywords:** botanical pesticides, deguelin, insects, rotenolone, rotenone, tephrosin, *Tephrosia**vogelii*

## Abstract

**Simple Summary:**

There is an increasing interest in botanical insecticides worldwide. Plants from the genus *Tephrosia* are rich in bioactive phytochemicals, particularly rotenoids which include rotenone, deguelin, rotenolone, and tephrosin. Rotenoids have strong insecticidal activities against a wider range of pests. However, there has been no treatise thus far focusing on *Tephrosia* as insecticidal plants. This article is intended to review phytochemicals produced by selected species, their insecticidal activities, and the current status on the use of *Tephrosia* as botanical insecticidal plants for insect pest control.

**Abstract:**

Synthetic insecticides are effective in controlling insect pests but can also harm nontarget organisms and the environment. During the last 40 years, there has been an increasing interest in alternative insecticides, particularly those derived from plants, commonly known as botanical insecticides. However, commercially available botanical insecticides remain limited. Rotenone is one of the earliest identified compounds and was used as fish poison and pest management. Due to its link with Parkinson disease, the use of rotenone was banned in many developed countries. Rotenone used to be isolated from *Derris* spp. and *Lonchocarpus* spp., and it can also be isolated from *Tephrosia* species. In this article, we present basic botanical information on selected *Tephrosia* species and their major compounds related to insecticidal activities and highlight the current use of extracts derived from some species, *Tephrosia vogelii* in particular, for control of insect pests in stored grains and crop production. The crude extracts contain multiple bioactive compounds, mainly rotenone, deguelin, rotenolone, and tephrosin, which act in either additive or synergistic fashion, resulting in effective control of insect pests. There are about 400 species in the genus *Tephrosia*, and species and even strains or variants vary greatly in these active compounds. We argue that a systematic evaluation of bioactive compounds in different species are needed, and species or strains with high insecticidal activities should be selected for use in the sustainable control of insect pests.

## 1. Introduction

Application of synthetic insecticides is the most common way of controlling insect pests in crop production [1]. The use of insecticides is credited for protecting crops from insect damage and enhancing crop productivity. It was estimated that without the use of pesticides, global food production loss would be 35–45% [2]. However, since the publication of *Silent Spring* [3], the application of synthetic insecticides has become one of the most controversial topics and remains the forefront regulatory issue in many countries. Insecticides are toxic, and indiscriminate use has led to the contamination of air, crop products, soil, and water; resurgence and resistance of insect pests; and toxicity to nontarget organisms [4,5,6].

An alternative to synthetic insecticides is botanical insecticides. Botanical insecticides are those consisting of dried, ground plant materials, crude plant extracts, or chemicals isolated from plants and used for managing insect pests [7]. Generally, plant secondary metabolites, like alkaloids, non-protein amino acids, steroids, phenols, flavonoids, glycosids, glucosinolates, quinones, tanins, and terpenoids are responsible for the protective action against insects. The use of botanical insecticides has a long history [7,8]. Pyrethrum was used in 400 BC. The first botanical insecticide was nicotine, dated back to the 17th century. In the mid of the 1800s, another botanical insecticide known as rotenone was introduced. Subsequently, sabadilla and other botanical insecticides were introduced [7,8]. In addition to the use of specific compounds isolated from plants, farmers across the world have used plants or plant extracts that have particularly potent defensive chemicals for controlling pests in the field and stored grains [9,10]. The modes of action of those plant extracts vary greatly including (a) confusants, phytochemicals that are unequivocal sign for insect pests to find their food source; (b) feeding deterrents, which are usually due to the activity of terpenes; (c) growth regulator activities, which can adversely affect insects’ growth and development and inhibit metamorphosis; (d) insecticidal activities resulting in the death on contact and/or by ingestion; and (e) repellents, some plants produce volatile compounds with bad odor or irritant, such as garlic and hot peppers to repel insect pests. Due to the occurrence of multiple active compounds in the extracts which may have different modes of action and could act additively or synergistically, the use of plant extracts would be more difficult to develop insecticide resistance [7,8,10] compared to the use of synthetic insecticides as they generally contain a single active compound with one mode of action [11,12]. Additionally, botanical insecticides are naturally occurring and act quickly and degraded rapidly, thus there is less pollution to the environment. Botanical insecticides have features of selectivity, low toxicity to livestock and natural enemies, with a few exception, low mammalian toxicity [7,8]. Botanical insecticides, particularly those plant-derived extracts from local farmers, are relatively cheaper in preparation and convenient in application with lower cost, due to the locally grown plants for local use in comparison to the use of synthetic ones [9,13].

Botanical insecticides have gained increasing attention as more than 20% of the research publications on insecticides are about botanical ones since the beginning of 1980 [8]. The most common botanical insecticides in the literature include pyrethrins, rotenone, sabadilla, ryania, nicotine, d-limonene, linalool, and neem [8]. However, commercially available botanical insecticides have been largely limited to pyrethroins and neem, to a lesser extent, limonene, chenopodium, capsicum oleorsin, and garlic oils [8]. Rotenone, one of the earliest identified compounds has not been widely used in insect management. Rotenone was initially isolated from ground roots or rhizomes of tropical legume *Derris* spp. and *Lonchocarpus* spp. The former produces up to 13% of rotenone, while the latter produces only about 5%. *Derris* spp. originated in tropical Asia, while *Lonchocarpus* spp. is native to western hemisphere. In addition to *Derris* and *Lonchocarpus* species, plants in the genus *Tephrosia* are also rich in rotenoid compounds. The content of rotenone in the pods of *Tephrosia vogelii* was 1.4% (dry weight) [14], and the amount of rotenoids varies with the *T. vogelii* variety and extractants [15] as the content of rotenoids in leaves varied from 0.65 to 4.25% [16]. *T. vogelii* is a fast-growing plant with large biomass. A yield of more than 14 metric tons of dried leaves and stems was harvested at Lewiston, North Carolina, the United States (U.S.), in a growing season [17]. Thus, plants in the genus *Tephrosia* have great potential for developing botanical insecticides. This article is intended to review general botanical characteristics of selected species, phytochemical production, and their potentials as botanical insecticides for insect pest control.

## 2. Common Species and Their Insecticidal Activities

The genus *Tephrosia* is a member of the family Leguminosae with about 400 species, and they are widely distributed in tropical, subtropical, and arid regions of the world [18,19,20]. It was reported that 50 species are native to equatorial Africa, of which 30 are found in Kenya; 70 are found in South Africa; 35 occur in India; and 30 are native of South America [21,22]. They are erect herbs or soft or woody shrubs with dense foliage grown to a height ranging from 0.5 to 4 m. They have the potential to restore soil fertility due to their ability to fix nitrogen [16,18]. Leaves are compound, leaflets are inverted lance-shaped or obovate, 7–15 cm long and 0.3–1 cm wide. Flowers are pea shape, white, purple or pinkish, 7 mm long, in a few-flowered, leaf-opposed, raceme-like clusters. Plants are self-pollinated and produce linear-long pods, 2.5–4.0 cm long and 3–4 mm wide. Seeds are ellipsoid, dark brown. Most species are diploid with a 2*n* chromosome number of 11. Species commonly associated with insecticidal activities include *Tephrosia candida,* (Roxb.) DC.; *Tephrosia elata*, Deflers; *Tephrosia purpurea* (L.) Pers.; *Tephrosia villosa* (L.) Pers.; *Tephrosia virginiana* (L.) Pers.; and *T. vogelii* Hook. f. (Table 1).

### 2.1. Tephrosia vogelii

Among the 400 species, *T. vogelii* is the most intensively studied one. It is a herb or small tree, native to tropical Africa and can attain a height of 2 to 3 m in a growing season of 5 to 7 months. Flower color may be white, purple, or red. It produces various compounds, such as flavonoid, steroid, and rotenoids (Table 1). Rotenoids mainly include four compounds (Figure 1): (a) Rotenone, it has a molecular formula C_23_H_22_O_6_, crystal with a melting point of 165 °C. Aphids are killed by 3 mg/kg of rotenone in solution and oral LD_50_ value in rats was about 132 mg/kg body weight [23]. (b) Deguelin, a derivative of rotenone with an empirical formula of C_23_H_22_O_6_, the same as rotenone, crystal, and melting point is 171 °C [24]. Its LD_50_ value to humans ranged from 10 to 100 g [25], but such values to insects are unknown. (c) Rotenolone, crystal with a chemical formula of C_23_H_22_O_7_. (d) Tephrosin, a nearly colorless rotenoid with a formula of C_23_H_22_O_7_ and melting point of 198 °C, and it is thought to be the oxidation product of deguelin [26]. The lethality values of both rotenolone and tephrosin are unknown. The content of rotenoids in *T. vogelii* leaves was higher than that in petal, stems, and roots, accounting for 80% to 90% of the total rotenoids [27].

**Table 1 insects-11-00721-t001:** Chemical constituents in some species of *Tephrosia* and their insecticidal activities.

Species	Class	Main Compound(s)	Mode of Action	Part(s) of Plant Used	Target Pests	Reference
*Tephrosia vogelii*	Rotenoid, flavonoid, and Steroid	Deguelin Tephrosin Rotenone Rotenolone α-toxicarol Elliptone Quercitin β-sitosterol Lanosterol Stigmasterol	Toxicity, larval toxicity, Antifeedant activity, Stomach or contact poison, Inhibition of oviposition, egg hatching and molting, Interference with growth and development, Interfering with the electron transport chain in mitochondria, Inhibition of cellular respiration and metabolism	Leaf Stem Root Seed Fruit coat	*Acanthoscelides obtectus* *Aedes aegypti* *Aphis fabae* *Aphthona whitfieldi* *Anopheles gambiae* *Bactrocera curcubitea* *Brevicoryne brassicae* *Callosobruchus chinensis* *Callosobruchus maculatus* *Caryedon serratus* *Crocidolomia pavonana* *Culex quinquefasciatus* *Dacus cucurbitae* *Diabrotica undecimpunctata* *Euschistus heros* *Megalurothrips sjostedti* *Monolepta* species *Pieris rapae* *Phyllotreta cruciferae*	[18,20,24,28,29,30,31,32,33,34,35,36,37,38,39,40,41,42,43,44,45,46,47,48]
*Tephrosia candida*	Flavonoid and rotenoid	Candidol Dehydrorotenone 12a-hydroxyrotenone Rotenone Tephrosin Amorpholone 6a,12,-dehydodeguelin 12a-hydroxy-β-toxicarol Deguelin α-toxicarol	Stomach or contact poison, Larval toxicity, Interference with growth and development, Antifeedant activity, Inhibiting the activity of NADH oxidoreductase in electron transporters	Stem Leaf Root Seed	*Aphis fabae* *Diaprepes abbreviates* *Spodoptera litura*	[20,28,31,49,50,51,52,53]
*Tephrosia elata*	Flavonoid and Rotenoid	Isopongaflavone Tephrosin (S)-elatadihydrochalcone Deguelin Rotenone	Antifeedant activity, Growth inhibition activity, Inhibiting the activity of NADH, Inhibition of cellular respiration and metabolism, Larval toxicity, Inhibition of oviposition	Seed Seedpod Root	*Aedes aegypti* *Eldana saccharina* *Maruca testulalis* *Spodoptera exempta*	[20,28,54,55,56]
*Tephrosia purpurea*	Rotenoid, flavonoid, and sterol	Tephrosin Rotenone Pongaglabol Semiglabrin Quercitin Rutin (+)-tephrorins A (+)-tephrorins B β-sitosterol	Interference with growth and development, Larval toxicity, Antifeedant activity, Inhibition of cellular respiration and metabolism, Anti-inflammatory and anti-cancer properties, Inhibition of ATP production	Whole plant	*Aedes aegypti* *Anopheles stephensi* *C* *ulex quinquefasiciatus* *Dysdercus cingulatus* *Odoiporus longicollis*	[20,28,57,58,59,60,61,62,63,64]
*Tephrosia villosa*	Rotenoid, sterol, and triterpenoid	Rotenone Dehydrorotenone Lupenone Stigmasterol	Larval toxicity, Stomach or contact poison, Interference with growth and development, Inhibition of oviposition Anti-parasitic	Whole plant	*Anopheles gambiae* *Culex quinquefasciatus* *Tenebrio molitor* *Spodoptera litura*	[20,47,65,66,67,68]
*Tephrosia virginiana*	Rotenoid	Rotenone Tephrosin Toxicarol	Larval toxicity, Interference with growth and development	Root	*Musca domestica*	[69,70]

Rotenoids belong to advanced members of the isoflavonoid group [71]. Their biosynthesis is based on shikimic/chorismic acid pathway, covering a group of plant natural products that is based on the rotoxen skeleton [71]. Rotenone was reported to be the most toxic of the rotenoids followed by deguelin [28]. Together, the four main compounds contribute greater than 95% of the toxicity [72]. Rotenone is a contact and ingestion compound [7]. Its mode of action involves the inhibition of the electron transport at the mitochondrial level, blocking phosphorylation of ADP (adenosine di-phosphate) to ATP (adenosine triphosphate), thereby inhibiting insect metabolism [28]. It is a selective, nonsystemic insecticide with contact and stomach action and secondary acaricidal activity [73]. Rotenone has been used for centuries as a selective fish poison [28] and also as an insecticide for controlling a wide range of arthropod pests including cucumber beetle, flea beetles, harlequin bug, leafhoppers, scales, spittlebugs, squash bugs, thrips, and some fruit worms [29]. Dried leaves of *T. vogelii* were used to protect stored legume seeds from damage by the bruchids [30].

### 2.2. Tephrosia candida

*Tephrosia candida*, commonly known as white hoarypea, is a perennial shrub, native to India. It grows up to 3.5 m. A distinct morphological characteristic of *T. candida* is that it produces greater biomass than *T. vogelii*. *T. candida* produces flavonoids including candidol, candidone, ovalichalcone, dehydrorotenone, candidin, and prongachin; and rotenoids, such as tephrosin and deguelin; and sterol (Table 1) [74]. Dehydrorotenone produced in stem and leaves is a stomach or contact poison and toxic to insects [20,75]. Tephrosin isolated from roots can interfere with insect growth and development [20,49]. Deguelin occurs mainly in roots and has antifeedant and growth inhibition activities, which are used for control of insects and nematodes [20,49]. A study conducted in Malawi showed that extracts derived from *T. candida* for control of aphid in common bean were less effective than those extracted from *T. vogelii*, which was explained by the fact that active compounds in *T. candida* were lower than *T. vogelii* [31].

*Tephrosia candida*, however, could be used as a cover crop for repellency of larval and adult diaprepes root weevil (Coleoptera: Curculionidae). This weevil, *Diaprepes abbreviates* (L.), is a major threat to the sustained profitability of citrus production in Florida and the Caribbean region. Adults were deterred from feeding and produced proportionally fewer eggs when caged with foliage of *T. candida* compared to foliage of *T. vogelii*, suggesting that the leaves of *T. candida* might contain antifeedants with activity toward adult diaprepes root weevil [76]. Additionally, feeding damage was observed by larvae on roots of *T. candida* and *T. vogelii*, but weight gain and survival of larvae fed with *T. candida* were greatly reduced compared to those fed with *T. vogelii*. These results implied that some compounds in root of *T. candida* are toxic to larvae. The authors suggested that phytochemicals responsible for the antifeedant and toxic properties of *T. candida* toward *D. abbreviates* are not shared by *T. vogelii,* and *T. candida* could be used as a cover crop in citrus field for control of diaprepes root weevil [76]. In fact, *T. candida* has been widely used in mixed cropping regimes and as a fallow crop in tropical production systems in Vietnam [77] and India [78] where it is valued for its contribution to soil fertility and simultaneously for repelling insect pest [49,50,79].

### 2.3. Tephrosia elata

*Tephrosia elata* is a short-lived, bushy perennial shrub that occurs in Kenya and produces various phytochemicals (Table 1). Two antifeedants: tephrosin and isopongaflavone were isolated from *T. elata* [54]. Isopongaflavone was reported to be effective against bean pod borer (*Maruca testulalis*) and African sugar-cane borer (*Eldana saccharina*). Tephrosin exhibited high activity against African armyworm (*Spodoptera exempta*). Isolated rotenone was able to control of African armyworm, African sugar-cane borer, and bean pod borer [55]. Roots, seeds, and seedpods contain flavonoid, which has antifeedant activity to insects, and interfere with insect growth and development [20]. Novel flavonoids: (s)-elatadihydrochalcone, obovatin methyl ether, and praecansone A were identified from seedpods. Seeds and roots contain rotenoids including deguelin and rotenone [55,56], which together showed larvicidal activity against third instar mosquito larvae of *Aedes aegypti* [55,56].

### 2.4. Tephrosia purpurea

*Tephrosia purpurea* is a highly branched suberect herbaceous perennial, about 1.5 m in height with spreading branches [80]. The plant grows abundantly in the upper Gangetic plains, and western Himalayas. More than 44 phytochemicals have been structurally identified from this species, which include rotenoids, flavanols, glycosides, isoflavones, sterols, and chalcones (Table 1) [81]. *T. purpurea* is known for its strong insecticidal efficacy. Sahayaraj [57] evaluated the potential of *T. purpurea* essential oil from stem and roots. Hexadecanoic acid was found to be the most abundant compound present. Essential oil showed strong repellent activity for males compared with females of banana stem weevil (*Odoiporus longicollis*), a serious pest of banana. The insecticidal efficacy was attributed to the presence of compounds like rotenone and hexadecanoic acid. *T. purpurea* leaf extract was able to control first to fourth instars larvae and pupae of *A. aegypti* [82]. Furthermore, the whole plant extract of *T. purpurea* was tested for its larvicidal activity against the larvae of *Culex quinquefasciatus*. The extract showed 100% mortality in very small doses suggesting its beneficial use in controlling the mosquito reproduction [58]. Winter season is more suitable for collection of plant materials due to the presence of high content of rotenone in this season [83].

### 2.5. Tephrosia villosa

*Tephrosia villosa* is a multibranched, perennial herb, up to 90 cm high, densely clothed with white, silky hair, found in India. Roots and seedpods produce flavonoids, including tephcalostaan, villosin, and tephrinone (Table 1). Whole plants contain rotenoids, dehydrorotenone, prenylated flavonone. The ethanol extract of roots, leaves, fruit, and twigs of *T. villosa* showed significant activity against southern house mosquito (*Culex quinquefasciatus*) larvae [65]. A defensin (TvD1) isolated from *T. villosa* showed inhibitory activities to mealworm (*Tenebrio molitor*) [84]. Plant defensin, a small, cationic, cysteine-rich broad-spectrum antimicrobial peptide, has four or five disulfide bridges and has been shown to be a component of the innate immunity system in plants. Over expression of defensin gene (*TvD1*) in tobacco exhibited strong activity against first and second instar larvae of taro caterpillar (*Spodoptera litura*), an important polyphagous insect attaching 44 families of economically important plants [66].

### 2.6. Tephrosia virginiana

*Tephrosia virginiana*, commonly known as devil’s shoestring as its roots are very long and stingy, such that they can be used for twine, occurs only in North America, ranging from Texas in the southwest to Florida in the east, north to Ontario, and west to Nebraska. Roots of this species were used as piscicide by Native Americans [85]. The roots contain rotenone, tephrosin, and toxicarol (Table 1) [69]. Root extracts are toxic to aphids, houseflies, potato beetle, fleas, and lice infecting dogs and poultry. Rotenone content in roots extracts harvested at the full-bloom stage was the highest compared to those harvested at dominant, emergence, and mature seed stages, and the extracts were highly toxic to houseflies (*Musca domestica* L.) [70]. Detailed information on phytochemicals of this species, however, have not been reported thus far.

## 3. The Use of *Tephrosia* Plants for Managing Insect Pests

The discussed *Tephrosia* species are rich in bioactive compounds and show insecticidal activities against different insect pests. Based on the current information, the use of *Tephrosia* plants for insect pest management can be summarized as the follows (Figure 2).

### 3.1. Commercial Formulation of Rotenone

Commercially, rotenone is generally extracted from the roots of cube plants (*Lonchocarpu utilis*) and barbasco (*Lonchocarpu urucu*) which was referred to as Cube resin as well as from derris plants (*Derris elliptica*). Rotenone is also extracted from *Tephrosia* spp and *Dalbergia paniculata* [28]. Although rotenone content in cube and derris plants is higher, about 5% in dried derri roots, the cultivation of these plants is difficult because of the liana type of growth and the labor involved in harvesting the small fibrous roots. On the other hand, most *Tephrosia* species have large biomass. A yield of over 14 metric tons of dried leaves and stems of *T. vogelii* could be obtained per hectare in the U.S. [17]. Regarding the content of rotenone in plant organs, leaflets contain 80% to 90% of rotenone [17]. Additionally, breeding effort by the U.S. Department of Agriculture (USDA) in the 1970s showed that some breeding lines could have rotenone content of more than 4.5% in leaflets [17]. Thus, extraction of rotenone from *T. vogelii* plants could be much easier and more convenient than from cube or derri plants.

Most commercial products of rotenone come from Central and South America [28]. Rotenone almost insoluble in water, and very soluble in many organic solvents, such as ethanol, acetone, chloroform, and ether [86]. Rotenone is unstable in light and air, and not environmentally persistent [87]. It degrades rapidly under natural conditions [24]. Rotenone powders lose much of their toxicity within weeks. Thus, its storage must be protected from air, light, and alkali, and the storage temperature should not exceed 25 °C. Solutions of rotenone in organic solvent, when exposed to light and air, become successively yellow, orange, and finally deep red due to oxidation [88]. Rotenone controls aster beetles, aphids, cabbage worms, cucumber beetles, Japanese beetles, and other insects [29]. Insects poisoned by rotenone experience a drop in oxygen consumption, respiratory depression, and ataxia, which lead to convulsions, paralysis, and death by respiratory arrest [73]. Due to its sensitive to air, light, and temperature, rotenone should be applied during cloudy sky or evening with appropriate dosage to maximizing insect control efficacy and reduce application frequency [10].

Rotenone was a registered pesticide in the U.S. under the Federal Insecticide Fungicide Rodenticide Act in 1947. Its formulations include crystalline preparations (about 95%), dust (0.75%), and emulsifiable solutions (about 50%). Rotenone is also formulated with other pesticides, such as pyrethrins, carbaryl, lindane, piperonyl butoxide, and others in products to control insects, mites, ticks, lice, spiders, and undesirable fish [28]. However, due to its potentially adverse impacts on aquatic ecosystems [89] and more recently the link to Parkinson’s disease in human beings [90,91], the U.S. has banned all uses of rotenone except as a piscicide since 2012, and the European Union (EU) began a phase out of rotenone in 2008 [92]. The Codex Alimentarius Commission in 2009 proposed to remove rotenone from the list of approved substances for plant protection, which was supported by Argentina, Japan, and Kenya but opposed by Australia, Brazil, Iran, Mexico, the Philippines, Thailand, U.S., and the International Federation of Organic Agriculture Movements (IFOAM) [93]. However, rotenone is still being used in China for controlling insect pests during vegetable production, such as head cabbage with the maximum residue limit set at 0.5 mg/kg [94].

Globally, the demand for natural pesticides is growing [7,8] due to the increasing interest in organically produced safe food. Policy changes in the U.S., EU, and some other countries about the safety data and maximum residue limits for synthetic pesticides as well as rotenone may have changed the commercial scope for botanical pesticides [95]. However, some countries, such as Brazil, China, and India have led the way in policy changes that could enabled more commercialization and use of botanical pesticides [8]. Nevertheless, the use of plant extracts for insect control has been a tradition, which remains strong across the African continent and some other regions [95].

### 3.2. Crude Extracts for Insect Control in Field Crop Production

Water, dilute liquid soap, or organic solvent-assisted extractions of bioactive compounds from plants continue although sophisticated procedures for extraction have been advanced. Crude or unrefined plant extracts are directly used for control of insect pests. Such practices remain in low input farming, particularly in Africa and even grow elsewhere like organic farming [96]. Insecticidal activities of the extracts could be due to the action of a single compound, additive or synergistic effects of several compounds. The combined effects have been referred to as phytocomplex [97]. A study conducted in Tanzania showed that extracts derived from *T. vogelii* significantly controlled aphid (*Aphis fabae* Scopoli), foliage beetle (*Ootheca mutabilis* (Schonherr) and *O. benigseni* Weise), and flower beetle (*Epicauta albovittata* Gestro and *Epicauta limbatipennis* Pic) in the common bean production field, and the application resulted in significantly higher yield than the control treatments [98]. Similarly, studies performed in Malawi and Tanzania showed that pest abundance (aphid, bean foliage beetle, and flower beetle) was lower when synthetic pesticides were used, while application of extracts derived from *T. vogelii* had relatively higher number of pests but lower than control treatments. More importantly, beneficial arthropod numbers were higher in plants treated with *T. vogelii* extract than those treated with synthetic insecticides, suggesting that *T. vogelii* extract had little effect on beneficial arthropods [96]. Such effects could be attributed to lower persistence of plant extracts and different modes of action [96]. To make the insecticide, dry powdered leaves were mixed with water containing 1% liquid soap at 10% w/v ratio for 24 h. Diluted solutions containing 1–2% of the extract were sprayed in the early evening to reduce exposure to sunlight and lessen effects on beneficial insects. Such practices have largely been used in southern and eastern Africa to control field pests rather than storage pests [99]. Previous studies indicate that *T. vogelii* is very effective in controlling a number of hard-to-kill field insects, including cucumber beetle, leafhoppers, squash bugs, flea beetles, harlequin bug, spittlebugs, thrips, scales, and some fruit worms [99]. Anjarwalla et al. [100] also reported the efficacy of extracts derived from *T. vogelii* in controlling bruchids in beans and cowpeas. Chemical analysis of *T. vogelii* indicated the presence of the rotenoids, including deguelin, tephrosin, and rotenone (with deguelin being the most abundant) [18,39]. These results demonstrated that the use of *T. vigelii* extracts to control insect pests can be as effective as synthetic insecticides in terms of crop yields, while conserving the non-target arthropods. Due to the locality of plant materials and convenience in extraction and application, the crude extracts can be more easily integrated in to agro-ecologically sustainable crop production systems [96].

### 3.3. Tephrosia as Cover Plant for Biocontrol of Insects and Soil Nitrogen Enrichment

Another role of *Tephrosia* plants is their repellency of insect pest. As mentioned above, adults of diaprepes root weevil were deterred from feeding of foliage on *T. candida* and produced fewer eggs, and *T. candida* could be used as a cover crop in citrus field for control of diaprepes root weevil [76]. Additionally, acetone extracts from leaves of *T. vogelii* had obvious antifeedant, inhibitory effect on growth and development and ovicidal activity, as well as acute stomach action [101]. The rotenoid compounds from *T. elata* showed significant antifeedant activity against *Maruca testulalis*, *Spodoptera. exempta*, and *Eldana sacchariana* [54]. Essential oil derived from *T. purpurea* showed stronger repellent activity against male banana stem weevils than the female ones [57]. The potentials of these species for repellency of insect pests deserve further investigation.

*Tephrosia* species are also widely used as cover crops by planting with rubber, oil palm, citrus, coffee, tea, coconut, and also annual crops [102]. Some species are reported to be green manure plants in agriculture and sometimes as a windbreak, contour hedge or shade plant because of the dense foliage and good root anchorage [103]. *T. noctiflora* is one of the *Tephrosia* species first being used as a green manure [103]. *T. candida* proved to be one of the most satisfactory green manures because it can flourish in poor soil for several years and has a dense foliage [103]. *T. candida* fallows alone can raise maize grain yield by 300% [104]. In South Rwanda, *T. vogelii* and *Cajanus cajan* were intercropped with *Sorghum bicolor* during the long rainy season. After sorghum was harvested, green manures were cut into small pieces and incorporated together with the sorghum residues. After four years of such intercropping, results showed that only *T. vogelii* led to a significant increase in yields. The yield of subsequently cropped beans increased by 25% compared to the control [105].

### 3.4. Tephrosia for Controlling Insect Pests in Stored Grains

*Tephrosia* plants have long been used for protecting grains from weevils [106]. Dried leaves have the potential to protect stored legume seeds from damage by bruchids in Southern Africa [107]. The insecticidal and repellent properties of *T. vogelii* were tested against *Sitophilus zeamais* Motschulsky (Coleoptera: Curculionidae) in stored maize grain. *T. vogelii* caused 85.0–93.7% insect mortality in 21 days. The mean lethal exposure times (LT_50_) to achieve 50% mortality varied from five to six days (7.5–10.0% *w*/*w*) to seven to eight days (2.5–5.0% *w*/*w*) [108].

Dry plant materials used for stored grains are generally recommended about 5% (*v*/*v*). However, due to the differences in the content of active compounds among varieties or even strains or variants, the application rates may vary. For example, the quantity of *T. vogelii* chemotype 1 (a strain or variant with higher insecticidal activity) used for control of adult bruchids could be much lower than that of chemotype 2 (a strain or variant with little insecticidal activity) [39]. The exposure of adult bruchids to chemotype 1 admixed with cowpeas led rapidly to higher levels of mortality than with chemotype 2. Interestingly, the exposure of bruchids to deguelin, the most abundant compound in the crude extract, supposedly less toxic than rotenone, caused a significantly higher mortality, while tephrosin was significantly less toxic than deguelin. The LC_50_ for tephrosin was calculated at approximately 200 mg/kg, whereas the LC_50_ values for rotenone, deguelin, sarcolobine, and toxicarol were below 10 mg/kg. The effect of obovatin 5-methyl ether against bruchids did not differ significantly from the control or the extract derived from chemotype 2 plants [39].

## 4. Concerns over the Use of *Tephrosia* Species as Botanical Insecticides

This review shows that *Tephrosia* plants are promising genetic resources for developing botanical insecticides. However, duo to the variable contents in active compounds among species or variants, along with incomplete understanding of their insecticidal activities, some concerns over the use of *Tephrosia* plants herein should be raised.

Not all species have anticipated insecticidal activities. As early as in the 1930s, Wilbaux [109] and Roark [16] suggested that only some species of *Tephrosia* might be the sources of rotenone. Irvine and Freyre [110] screened 16 *Tephrosia* species, 14 were contained some rotenoids, five contained rotenoids in their leaves, and *T. vogelii* had the highest leaf rotenoid content, ranging from 0.65% to 4.25%. Additionally, distinct chemotypes or variants occur in *T. vogelii*. Chemical analysis of plant material across Malawi identified two distinct chemotypes, one containing rotenoids for their biological activity against insects [111,112] and the other was characterized by flavones, flavanones, and flavonols [18]. Subsequent bioassays revealed that insecticidal activities of chemotype 1 were due to the presence of rotenoids, including deguelin, dehydrodeguelin, rotenone, and tephrosin; while the flavonoids in chemotype 2 were inactive [39] and had little active against insects. It was reported that about 25% of the plants of *T. vogelii* grown in Malawi belonged to the chemotype 2. Thus, the use of wrong species or chemotypes, such as chemotype 2 could result in ineffective in pest management as mentioned by Stevenson and Belmain [95].

Insecticidal activities may not completely rely on rotenone. In the same study [112], chemotype 1 contained deguelin as the major rotenoid along with tephrosin, and rotenone as a minor component. As mentioned above, extracts from chemotype 1 plants showed insecticidal activities, meaning deguelin plays a critical role in control of insect pests. This finding is important. Currently, it is rotenone that has been banned in several countries [92], but there has been no documentation about any regulations of deguelin. Interestingly, it has been known that plants containing high deguelin have been used as an anthelmintic agent in certain regions of China [113] and as a traditional Thai medicine to treat hepatitis and hepatic dysfunction [114]. Recently, deguelin has been shown to have potent anticancer activity against multiple cancer types and can impede carcinogenesis by enhancing cell apoptosis and preventing malignant transformation and tumor proliferation [115]. *Tephrosia* species or chemotypes with high deguelin contents but low levels of rotenone could be ideal plant materials for developing botanical insecticides. These results may imply that future chemical evaluation of *Tephrosia* species or chemotypes should focus more on deguelin. A study conducted in West Java, Indonesia showed that acetone extracts derived from leaf samples collected from seven geographical locations varied in LC_50_ (ranging from 0.137% to 0.371%) against cabbage head caterpillar (*Crocidolomia pavonana*). More importantly, compounds other than rotenone are also responsible for the insecticidal activity [116]. Early studies also showed that the leaflets of *T. vogelii* contained more deguelin than rotenone, the reverse was general found in the petioles, stems, and roots [17]. One variety contained deguelin but no rotenone [27]. Varietal or strain variation in active compounds should be isolated individually and labelled as different chemotypes or specific strain. They should be produced in isolation for seed production or simply propagated by vegetative means to maintain the genetic identity for further evaluation and potentially released as new cultivars for commercial production.

A major challenge in production of *T. vogelii* is poor seed production [17], which has affected breeding effort on developing new cultivars and large-scale production of selected desirable chemotypes or strains [17]. With the advance of plant tissue culture or microprogation [117], this problem can be easily resolved. Micropropagation through shoot culture using existing meristems can produce a large quantity of plantlets without somaclonal variation [118]. Thus, the availability of genetic identical plantlets can be used for commercial production. Due to their rapid growth characteristic, a large quantity of biomass can be used for extracting bioactive compounds for pest management.

Precautions should be taken when handling *Tephrosa* extracts as some species or variants contain high rotenone, and rotenone has been linked with Parkinson’s disease. Additionally, the extracts may have other potential hazardous effects to nontarget organisms even though they are much safer than commercial rotenone insecticide. The precautions include safe procedures for extraction, appropriate labeling of the extracted products, methods for storage and transportation, and safe methods for field applications at right time and correct doses.

## 5. Conclusions and Future Outlook

Several species of *Tephrosia* have long been recognized as insecticidal plants. Studies of these plants over the last 60 years have resulted in some important findings. (a) Species and varieties differ significantly in rotenoid contents. The USDA program conducted from the 1960s to 1970s clearly showed that rotenoid content varied from 0.65% to 4.25%, and hybridization and selection could further increase rotenoid content [17]. This finding suggests that rotenoid content can be improved through breeding. With the advance of molecular marker tools and omics technologies, we believe that rotenoid contents can be significantly enhanced by modern breeding technologies. (b) Chemotypes or variants occur within varieties of *T. vogelii*, individual plants may differ significantly in rotenone, deguelin, rotenolone, and tephrosin contents (Belmain et al., 2012). (c) Deguelin could be an important compound against insect pests. Current data indicate that deguelin could be equal to or more important than rotenone in pest management. Deguelin has been widely used as pharmaceutical compounds [115], its use as an insecticide may be environmentally more benign than rotenone. Thus, the potential of deguelin deserves future investigation. (d) Progress made on the use of *T. vogelii* in control of insect pests in Africa is encouraging. The use of crude extracts can control insects under the threshold level, protect other arthropods, and result in high crop yield. Such work demonstrates the effectiveness of a phytocomplex for sustainable pest control in contrast to some negative effects with the use of synthetic insecticides.

The progress made in *T. vogelii* calls for action to explore the rich genetic resources of *Tephrosia*. (a) Bioactive compounds including rotenone, deguelin, rotenolone, and tephrosin in different organs of each species should be analyzed. Based on the results, species with high levels of active compounds and insecticidal activities should be further screened for individual rotenoid contents among variants or chemotypes. (b) Chemotypes or variants with high contents of the compounds should be propagated vegetatively though in vitro shoot culture to maintain their genetic identity and fidelity. (c) These chemotypes or variants should be used as parents for hybridization to produce segregated populations for selecting progenies with even higher bioactive compounds. The developed breeding lines should be evaluated in different regions for consistence in performance, and new cultivars with high levels of a compound or a group of compounds should be released. (d) Crude extracts should be extracted from the developed cultivars, and they should be evaluated and used for control of insect pests. (e) Attention should be given to deguelin, it should be evaluated for insect control and effects on nontarget organisms and the environment. It is anticipated that deguelin could be an important compound for management of insect pests in the near future.

## Figures and Tables

**Figure 1 insects-11-00721-f001:**
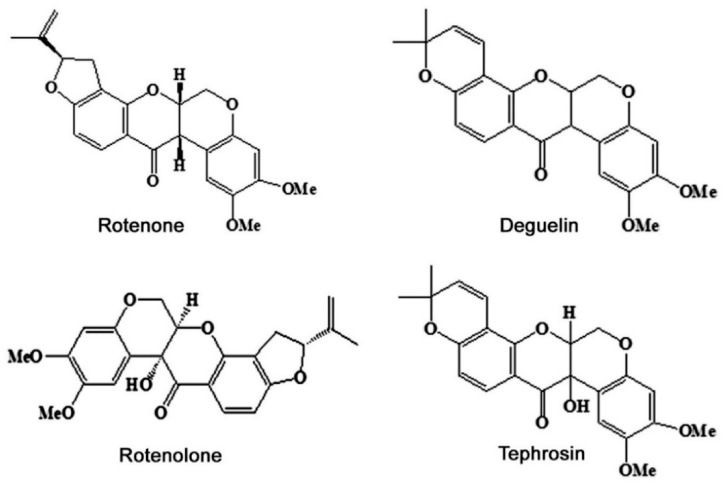
Major chemical constituents in *Tephrosia vogelii* plants.

**Figure 2 insects-11-00721-f002:**
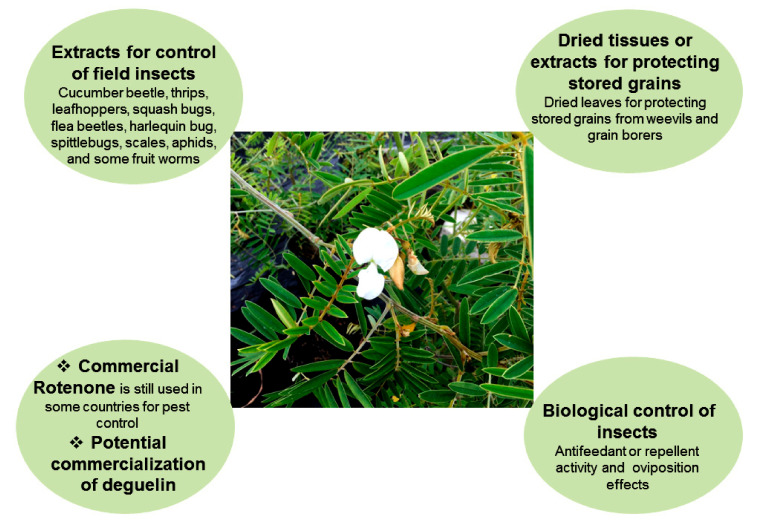
The use of *Tephrosia* plants for pest management.

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
