# Peer review of "Plants in the Genus Tephrosia: Valuable Resources for Botanical Insecticides"

_insects, 2020, doi:10.3390/insects11100721_

Round 1

Reviewer 1 Report

I attach my opinions

Sincerely yours

Author Response

Thank you so much for reviewing our manuscript and for your constructive comments and suggestions. In this revision, we have appropriately addressed all the concerns listed in your peer-review-8987285.v2 file, which can be summarized as follows:

  1. All requested bibliographic citations have been provided.
  2. Some questionable terms or sentences have been revised.
  3. Explanations for some sentences or information have been added.
  4. Information unrelated to the scope of this review has been deleted.
  5. Table 1 has been reorganized and revised by listing insect pests under the target pests.
  6. Figure 1 has been modified by deleting some information unrelated to the scope of the review.
  7. Cited references and references under the Reference section have been reorganized.

Reviewer 2 Report

The manuscript makes a comprehensive review on the genus of plants Tephrosia, in view of the need to expand the knowledge on insecticide botanicals that can help and control pests.

Plant species of the genus Tephrosia contain many phytochemicals that have bioactivity, especially rotenoids that include rotenone, deguelin, rotenolone, and tephrosin.

Known rotenoids have insecticidal activities against a wide variety of pests.

This manuscript aims to review the general morphological characteristics of the species, the production of phytochemicals and the current status of the use of Tephrosia as insecticidal plants that can act and collaborate in the control of insect pests.

In my opinion, the review is well structured and can be published as it will allow a significant advance of knowledge in this area. I am in favor of publication.

Author Response

Thank you so much for reviewing our manuscript and for your positive comments. We have revised the manuscript and believe that the quality of this manuscript is further improved.

Reviewer 3 Report

The manuscript is an interesting overview of genus Tephrosia as a potential source of insecticidal components in particular rotenoids. The review is well organized and many aspects are deeply discussed; however, rotenoids and their botanical sources show some limitations regarding their useful insecticidal activity and their safety towards mammalian and humans. For these reasons, I ask the authors to better discuss the potential toxicity towards human of the rotenoids and their botanical sources to complete the information reported in this review

Author Response

Thank you so much for reviewing our manuscript and for your constructive comments and suggestions. In this revision, we have addressed your concern by adding more information in the third paragraph under “4. Concerns over the use of Tephrosia species as botanical insecticides”.